# Osteogenic Effect of a Bioactive Calcium Alkali Phosphate Bone Substitute in Humans

**DOI:** 10.3390/bioengineering10121408

**Published:** 2023-12-11

**Authors:** Christine Knabe, Doaa Adel-Khattab, Mohamed Rezk, Jia Cheng, Georg Berger, Renate Gildenhaar, Janka Wilbig, Jens Günster, Alexander Rack, Max Heiland, Tom Knauf, Michael Stiller

**Affiliations:** 1Department of Experimental Orofacial Medicine, Philipps-University Marburg, 35039 Marburg, Germanymzakaria_rezk@yahoo.com (M.R.); ccheng.jiajia@gmail.com (J.C.); tom.knauf@med.uni-marburg.de (T.K.); stiller@implant-consult.de (M.S.); 2Department of Oral and Maxillofacial Surgery, Charité University Medical Center Berlin (Charité-Universitätsmedizin Berlin), Corporate Member of Freie Universität Berlin, Humboldt-Universität zu Berlin, and Berlin Institute of Health, 13353 Berlin, Germany; max.heiland@charite.de; 3Department of Oral Medicine, Periodontology and Diagnosis, Faculty of Dentistry Ain Shams University, Cairo 11566, Egypt; 4Division “Advanced Multi-Materials Processing”, Federal Institute for Materials Research and Testing, 12203 Berlin, Germanyrenate.gildenhaar@bam.de (R.G.); janka.wilbig@bam.de (J.W.); jens.guenster@bam.de (J.G.); 5Structure of Materials Group, European Synchrotron Radiation Facility, 38043 Grenoble, France; 6Department of Traumatology, Philipps-University Marburg, 35043 Marburg, Germany

**Keywords:** bioceramics, calcium alkali orthophosphate materials, bioactive bone grafting material, bioactivity, osteogenesis, bone regeneration, sinus floor augmentation, silicon release

## Abstract

(1) Background: The desire to avoid autograft harvesting in implant dentistry has prompted an ever-increasing quest for bioceramic bone substitutes, which stimulate osteogenesis while resorbing in a timely fashion. Consequently, a highly bioactive silicon containing calcium alkali orthophosphate (Si-CAP) material was created, which previously was shown to induce greater bone cell maturation and bone neo-formation than β-tricalcium phosphate (β-TCP) in vivo as well as in vitro. Our study tested the hypothesis that the enhanced effect on bone cell function in vitro and in sheep in vivo would lead to more copious bone neoformation in patients following sinus floor augmentation (SFA) employing Si-CAP when compared to β-TCP. (2) Methods: The effects of Si-CAP on osteogenesis and Si-CAP resorbability were evaluated in biopsies harvested from 38 patients six months after SFA in comparison to β-TCP employing undecalcified histology, histomorphometry, and immunohistochemical analysis of osteogenic marker expression. (3) Results: Si-CAP as well as β-TCP supported matrix mineralization and bone formation. Apically furthest away from the original bone tissue, Si-CAP induced significantly higher bone formation, bone-bonding (bone-bioceramic contact), and granule resorption than β-TCP. This was in conjunction with a higher expression of osteogenic markers. (4) Conclusions: Si-CAP induced higher and more advanced bone formation and resorbability than β-TCP, while β-TCP’s remarkable osteoconductivity has been widely demonstrated. Hence, Si-CAP constitutes a well-suited bioactive graft choice for SFA in the clinical arena.

## 1. Introduction

Over the last three decades, maxillary sinus floor grafting has become a widely used alveolar ridge augmentation procedure preceding dental implant placement in the atrophic posterior maxilla [1,2,3,4,5,6,7,8]. Bioactive calcium phosphate (CaP)-based glasses and ceramics are bone-bonding and enhance bone cell function and osseous tissue formation due to surface-mediated effects and effects mediated by ion dissolution products [4,9,10]. Over the last 25 years, resorbable bioactive CaP-based bone substitute materials such as β-tricalcium phosphate (β-TCP) have been increasingly explored for sinus floor augmentation (SFA), so as to avoid autogenous bone harvesting and donor site morbidity [7,11]. The clinical success rates for SFA achieved with β-TCP show that β-TCP is a well-suited alternative graft choice to autologous bone, which is widely regarded as the gold standard [5,6,7,8,12,13,14,15,16]. With respect to β-TCP, on the basis of histologic examination of tissue sampled at dental implant placement, several authors demonstrated that after SFA, β-TCP resorbed within 1–2 years [1,7,8,17,18]. This resulted in an increasing quest for bioactive bioceramic biodegradable bone substitute materials that stimulate osteogenesis and biodegrade rapidly in the newly formed bone, leading to bone regeneration and replacement by fully functional osseous tissue which is important for inserting dental implants into these grafted sites [19]. Hence, optimizing the osteogenic efficacy of bioactive CaP bone substitutes has been the topic of extensive research with the goal of achieving more copious bone tissue formation in less time, which in turn enables earlier dental implant placement and shortened treatment durations [4,19]. An ideal bone grafting material should attract osteoprogenitor cells and accelerate their osteoblastic cell differentiation into mature osteoblasts, which elaborate the extracellular bone matrix, elicit its mineralization, and thus stimulate bone neoformation at their surface in conjunction with an increased resorption rate, leading to balanced rapid bone formation and rapid resorption within the newly formed bone [4,9]. This has initiated the creation of bioactive, rapidly resorbable calcium alkali orthophosphates (CAPs) with amorphous phases and the crystalline phase Ca_2_KNa(PO_4_)_7_. These CAPs display a greater solubility and resorbability than β-TCP [4,20,21,22,23,24,25].

In previous studies, we demonstrated that a silica-doped calcium alkali phosphate GB9 (Si-CAP) exhibited a higher enhancement effect on bone cell differentiation, bone matrix maturation, and tissue formation both in vitro and in large animal ovine models in the context of alveolar ridge augmentation and SFA than the widely clinically used bone substitutes β-TCP, bioglass 45S5 (BG 45S5), or other CAPs [4,22,23,24,25]. This effect was related to calcium uptake at the top layer of the Si-CAP bioceramic, to silicon-ion release, and increased serum protein adsorption of fibronectin, as well as simultaneous enhanced activation of intracellular signaling that modulates bone cell differentiation as well as survival [22,23,24,25]. This was in addition to displaying a greater resorbability than β-TCP, BG45S5, and other CAPs [4]. Collectively, these findings led to FDA approval.

Furthermore, histologic and histomorphometric analysis are important tools for studying bone regeneration [26]. Detailed histological analysis and histomorphometric measurements regarding the bone repair process are critical for demonstrating therapeutic efficacy and addressing questions regarding cell and tissue responses to endosseous implant materials during bone neoformation. These analyses can be combined with molecular and radiological data, generating a comprehensive dataset of outcome findings corroborating each other [27]. As such, detailed immunohistochemical and histologic analysis of the bone cell and tissue responses to bone substitutes after implantation in vivo can contribute significantly to generating the required knowledge base for the evidence-based application of these bone substitutes in humans and for successful translation to the clinic. In this context, we previously developed a hard tissue histologic technique that enables visualizing osteoblasts that actively form and secrete an osseous matrix and induce its mineralization at the bone–bioceramic interface as well as in the pores of degrading bioactive bone substitutes at varying time points after the grafting procedure [23,28,29]. This technique also allows us to visualize active osteoblasts and bone matrix mineralization in the osseous tissue that forms in response to these bioactive bone substitutes. Consequently, this methodology allows us to validate the actual bioactive properties of bone substitutes in vivo, i.e., their capability to attract osteoprogenitor cells and to induce osteoblast differentiation at their surface in conjunction with bone matrix mineralization in preclinical large animal models and patients without removal of the bioceramic from the bone tissue sections, which is required in routine decalcified paraffin histology [3,4,28].

The objective of the current study was to correlate the results of previous cell culture and large animal studies on the osteogenic capacity of Si-CAP versus that of β-TCP, with the histomorphometric and immunohistochemical findings regarding the effect of Si-CAP on bone neoformation six months after sinus floor grafting in humans in comparison to β-TCP. These previous studies had analyzed the effect of Si-CAP on intracellular signaling, and osteoblast differentiation in vitro, and on bone cell and tissue maturation and formation in clinically representative large animal models when compared to β-TCP. Consequently, our study tested the hypothesis that greater osteoblast differentiation and bone tissue formation in vitro and in sheep in vivo would translate into enhanced bone neoformation in patients six months after sinus floor grafting utilizing Si-CAP granules when compared to β-tricalcium phosphate granules.

## 2. Materials and Methods

### 2.1. Bone Grafting Materials

Test materials included the following: first, glassy crystalline Si-CAP granules which displayed the main crystalline phase Ca_2_KNa(PO_4_)_7_ and a glassy portion (4%) containing sodium magnesium silicate (granule size from 1000 to 2000 µm; porosity 75%, Osseolive^®^, Curasan Ltd., Hesse, Germany, Kleinostheim, FRG). The granules exhibited an open cellular microarchitecture (Figure 1a) resembling that of cancellous bone. Synthesis, fabrication, and characterization of this material has been described in detail previously [25,28]. In brief, for the fabrication of Si-CAP, a mixture of CaO, P_2_O_5_, Na_2_O, K_2_O, and MgO was melted and then quenched. Si-CAP was doped with 4% sodium magnesium silicate. For the fabrication of Si-CAP granules, first, cancellous scaffolds with an open cellular structure were prepared by employing the Schwartzwalder Somers replica technique that entailed utilizing highly porous combustible polyurethane templates. The respective powder materials were used for producing a homogenous water-based slurry, to which the doping component was added. The open cellular polyurethane foam was coated with the slurry in order to produce highly porous scaffolds. The coated polyurethane template was subsequently dried at 50 °C followed by sintering at 1000 °C, which entailed pyrolysis of the foam. The ceramic scaffolds were then milled (Pulverisette, Fritsch, Idar-Oberstein, Germany) followed by a sieving procedure to produce Si-CAP with a size of 1000–2000 µm [28]. Secondly, synthetic pure-phase β-TCP granules displaying a granule size of 700 to 1400 µm and a 70% porosity (denominated β-TCP; CEROS^®^ β-TCP granules, Mathys Medical, Bettlach, Switzerland) (Figure 1b) served as the reference material. β-TCP granules possess interconnected macropores (100–500 µm in size) and a fraction of micropores (1–10 µm in size). Fabrication and material characterization of this bone grafting material has been described in detail elsewhere [6]. In short, the β-TCP granules were produced utilizing a similar replica manufacturing process as described for Si-CAP, which generated β-TCP scaffolds with an interconnected porosity featuring macropores 100–500 µm in size. To this end, a polymer template was coated with a pure phase β-TCP slurry, followed by sintering at 1000 °C–1300 °C. The obtained TCP scaffolds were then milled followed by sieving to produce β-TCP granules with a granule size of 700 to 1400 µm. Both materials were cleared by the FDA and received the CE mark.

### 2.2. Patient Selection and Patient Clinical Data

A total of 23 female and 15 male patients (mean age: 60 ± 9 years; range: 40–76 years) were included in this study. The inclusion criteria were as follows: partial edentulism in the premolar and molar region requiring sinus floor grafting to facilitate dental implant surgery in the posterior maxilla due to the height of the atrophic alveolar crest being below 3 mm; width of the alveolar crest of at least 6 mm, so as to facilitate easy biopsy sampling; good oral health; being non-smokers; good general health with the absence of chronic conditions; absence of active periodontitis; absence of pathological conditions of the maxillary sinuses. The exclusion criteria were as follows: compromised health (ASA (III or IV)—American Society of Anesthesiology); smoking; chronic conditions such as diabetes, obesity, and chronic inflammatory conditions; immune suppression; active periodontitis; poor oral hygiene; pathologic conditions of the maxillary sinuses and Schneiderian membrane. The study was approved by the Freiburg Ethics Commission International (code ZD-MA-MS-2013-1). Written informed consent was obtained from all participants, who had been fully informed regarding the procedures such as the surgery and biomaterials, i.e., dental implants as well as bone substitutes (registry number DRKS00007538).

### 2.3. Cone Beam Computed Tomography and Panoramic Radiographs

Cone beam computed tomography (CBCT) (KaVo-3D-eXam^®^, KaVo Dental GmbH, Biberach, Germany) was utilized for planning the procedures, for assessing sinus floor anatomy and bone volume preoperatively, as well as for excluding pathological conditions of the maxillary sinuses. Furthermore, acquisition of routine panoramic radiographs was performed pre- and postoperatively, 6 months after sinus floor grafting prior to and immediately after dental implant placement.

### 2.4. Surgical Interventions and Biopsy Sampling

Sinus floor augmentation was carried out under local anesthesia employing the lateral window approach according to Tatum (1986) by the same experienced surgeon in all patients [30]. After mixing with venous blood, Si-CAP granules or β-TCP granules were used for filling the void created between the osseous sinus floor and the elevated Schneiderian membrane. The detailed surgical procedures and perioperative medication have been described elsewhere [5,6]. Six months after SFA patients received the implants, and bone biopsy sampling was performed utilizing a trephine burr. The biopsies were approximately 9 mm long and 2.5 mm in diameter. These samples were then subjected to histomorphometric and immunohistochemical analysis. The samples contained the grafted region as well as the residual native crest (Figure 2), which was excluded from the histomorphometric analysis.

### 2.5. Histologic, Histomorphometric, and Immunohistochemical Analyses

The bone biopsy samples were processed for hard tissue histology using a technique which allowed immunohistochemical analysis of undecalcified hard tissue sections as outlined in detail elsewhere [5,29]. In brief, a Leitz 1600 sawing microtome (Leica, Wetzlar, Germany) was used for cutting 50 µm thick sections after resin embedding. Sections were then ground and polished. Deacrylation of the sections was followed by immunohistochemical staining with primary mouse monoclonal antibodies specific to osteocalcin (OCN) (Abcam, Cambridge, UK), alkaline phosphatase (ALP) (Sigma, Darmstadt, Germany), and rabbit polyclonal antibodies against collagen type I (Col I) (LF-39, NIH)) and bone sialoprotein (BSP) (LF-84, NIH, Bethesda, Rockville, MD, USA). Mayer’s hematoxylin was utilized for counterstaining. Non-immunized mouse and rabbit IgG (PP54 and PP64) (Millipore, Billerica, MA, USA) served as negative controls. Histomorphometric analysis was carried out on a pair of sections 150 µm apart. A light microscope (BX-63), a digital camera, and CellSenseTM software version V3.2 (Olympus, Germany) were employed, and a square ROI 4 mm^2^ in size was defined in two regions of each section: apically underneath the Schneiderian membrane and secondly in the central region of the cylindrical biopsy at a distance of 3 mm from the native bone of the alveolar crest (Figure 2). The bone area fraction, the graft material area fraction, and the bone-bioceramic contact were determined in both ROIs in order to characterize bone formation, the biodegradability of the bone grafting materials, and their bone bonding behavior, as described previously [6,23]. Data from each pair of sections were averaged. Furthermore, the immunohistochemically stained sections were subjected to semi-quantitative analysis as described previously [23,31,32]. Analysis of the stained sections was carried out by two experienced investigators who were blinded to the staining. The cellular components analyzed were osteoblasts, osteocytes, and fibroblasts, and the matrix components analyzed were trabecular bone, osteoid seams, bone marrow spaces, and fibrous matrices. A scoring system was utilized for quantifying the amount of staining recorded for the different osteogenic markers: a score of (+++[=5]), (++[=4]), and (+[=2]) indicated generalized strong, moderate, or mild staining, and a score of (+++[=4]), (++[=3]), and (+[=1]) indicated localized strong, moderate, or mild staining. A score of (0) was used for no staining. The average score of the 19 sections per osteogenic marker was calculated. An average score of 3.5–5 was assessed as a strong expression of a respective marker in a given cellular or matrix component, and an average score of (2.3–3.4), (1–2.2), and (0.1–0.9) was evaluated as moderate, mild, and minimal expression. In addition, high-resolution synchrotron microtomography was performed on various biopsies as described previously [15], in order to obtain 3D visualization of the newly formed bone tissue and the residual ceramic bone substitute material in the biopsies sampled from the grafted sinus six months after grafting.

### 2.6. Statistical Analysis

The histomorphometric data were expressed as mean ± standard deviation, and the Mann–Whitney U test was employed for statistical analysis of non-parametric data (StatsDirect software; version 3.0). Statistical significance was considered achieved at *p* < 0.05.

## 3. Results

### 3.1. Clinical Intraoperative and Postoperative Findings

Patient age and gender are listed in Table 1. Following sinus floor grafting surgery, neither postoperative complications nor inflammatory reactions were noted in any patients. In all patients, sufficient bone levels were obtained for achieving adequate primary stability at dental implant surgery six months after sinus floor grafting.

### 3.2. Radiological Findings

Panoramic radiographs and CBCTs did not show any pathological changes in the grafted sinuses or the neighboring tissues such as the Schneiderian membrane both postoperatively after SFA and six months later at dental implant surgery. At this time point, sinus floors augmented Si-CAP displayed excellent osseous tissue formation radiographically with extensive substitution of the Si-CAP bone substitute by new bone without any inflammatory pathological tissue reactions. CBCTs of sinus floors grafted with β-TCP, in contrast, revealed a higher amount of residual β-TCP bone substitute material six months after grafting of the sinus floor (Figure 3a–d).

### 3.3. Results of Histologic, Histomorphometric, and Immunohistochemical Analyses

Histological analysis revealed that both bone substitutes supported excellent bone neoformation and matrix mineralization with active progression from the sinus floor in a cranial direction six months after sinus floor grafting (Figure 4, Figure 5 and Figure 6). Bone formation was preceded by a mesenchyme rich in osteoprogenitor cells exhibiting positive expression of OCN, Col I, ALP, and BSP in the osseous tissue components. The findings of the histomorphometric analysis are illustrated in Figure 4. The apical region of Si-CAP biopsy samples is at the greatest distance from the native osseous tissue of the sinus floor, a significantly higher bone area fraction (mean 33.91%, *p* < 0.0005), i.e., more copious bone formation and bone bonding, that is, bone-bioceramic contact (mean 53.48%, *p* < 0.0002), were recorded than in biopsies sampled from sinus floors grafted with β-TCP (mean bone area fraction 9.38%, mean bone-bioceramic contact 15.89%, Figure 4b). This was accompanied by a significantly lower grafting material area fraction being observed apically in the Si-CAP specimens (mean 17.26%) compared to the β-TCP group (mean 29.4%, *p* < 0.0012) (Figure 4b) indicating greater biodegradability for Si-CAP. Consequently, in the apical ROI, bone formation, bone-bonding, and graft material resorption were significantly greater, i.e., more advanced, with Si-CAP than with β-TCP (Figure 4b).

In the central ROI, this was true to a lesser extent, due to the differences lacking statistical significance (Figure 4a). In addition, apically, significantly higher BSP expression was noted in the bone matrix, as well as significantly higher expression of OCN and Col I in the osteoid and of OCN in the fibrous matrix of the osteogenic mesenchyme, in combination with higher expression of Col I, BSP, and OCN in osteoblasts and osteocytes in the Si-CAP samples in comparison to β-TCP specimens (Table 2).

Centrally, significantly greater Col I and BSP expression was observed in the bone matrix, and in the osteoid (Col I), significantly greater OCN and BSP expression was observed in fibroblastic cells of the osteogenic mesenchyme as well as stronger (though statistically not significant) ALP, BSP, OCN, and Col I expression in osteoblasts when comparing Si-CAP biopsies to β-TCP samples. Moreover, excellent trabecular bone formation in the grafted sinus floor was demonstrated by histological analysis, which also showed that highly leached and degraded fragments of the Si-CAP granules were embedded in the regenerated osseous trabeculae and that extensive bone ingrowth into the pores of these granule residues had occurred resulting in excellent bone bonding behavior (Figure 5d,e—yellow arrows). Remodeling of the original cancellous alveolar bone microanatomy of the maxilla was more advanced in Si-CAP sites. In addition, at the degrading Si-CAP granule surface, active matrix mineralization and osseous tissue formation with strong OCN expression were present in the apical region of the biopsies in close proximity to the bordering sinus mucosa (Figure 5d,f–h, green arrows). As such, progressing bone neoformation occurred in tandem with continuously progressing degradation of the residual Si-CAP granules (Figure 5d,h—blue arrows), which were replaced by the new bone tissue (black arrows in Figure 5e–h). In addition, copious capillary formation was visible in these pores of the residual Si-CAP granules (Figure 5i—orange arrows). Moreover, synchrotron microtomography revealed a larger amount of residual bioceramic material in β-TCP biopsy samples (Figure 6b) than in Si-CAP biopsy specimens (Figure 5b), which furthermore looked more like mature bone (Figure 5a) at the macroscopic level compared to β-TCP samples (Figure 6a). β-TCP granules exhibited a scalloped morphology with varying degrees of degradation and a considerably larger size (Figure 6d,e) than the residual Si-CAP granules indicating a lower biodegradability.

## 4. Discussion

Translation of novel biomaterials to the clinic should involve detailed characterization of the cell and tissue responses to these biomaterials in clinically relevant animal models followed by clinical studies [33,34]. In this context, the evidence-based use of bone grafting materials should entail understanding the cellular and molecular events occurring at the bone–biomaterial interface in addition to demonstrating the superiority of novel materials over existing materials or therapeutic concepts based on comprehensive data from clinical studies. As such, the current study aimed at generating detailed histological data regarding the bone regeneration process in patients six months after sinus floor grafting with the Si-CAP grafting material as compared to β-TCP in order to facilitate evidence-based translation of Si-CAP to the clinical arena.

Our first findings in 19 patients, which showed enhanced osteogenic marker expression, significantly greater bone neoformation, bone bonding, and biodegradability for Si-CAP apically, are consistent with findings of former in vitro osteoblast and in vivo sheep studies. In these studies, a greater enhancement effect of the glassy crystalline material Si-CAP on osteogenesis in vitro as well as osseous tissue maturation and formation in vivo was demonstrated when compared to the clinically widely utilized bone substitutes β-TCP and BG 45S5 [4,22,23,25]. This was in the context of bone regeneration of the ovine sinus floor and of critical size defects in the ovine mandible with the GBR (Guided Bone Regeneration) technique, as well as in the ovine scapula [4,25]. In the present study, apically greater BSP expression, which is indicative of active matrix mineralization, was noted in the mineralized bone matrix in Si-CAP biopsies in comparison to the β-TCP group. This was in addition to greater OCN expression in the osteoid, indicating active bone apposition. In sheep, Si-CAP granules, which were located apically in close proximity to the Schneiderian membrane at a large distance from the sinus floor and thus native bone, induced matrix mineralization and bone formation at their surface [4]. Thus, histologically the same phenomena, which previously had been observed in sheep, were present in the human Si-CAP biopsies (Figure 5). This effect was shown to be related to calcium uptake at the top layer of the Si-CAP bioceramic, silicon-ion release and simultaneous enhanced upregulation of the ERK differentiation, the PI3K cell survival and the alternate p38 pathways, as well as enhanced fibronectin serum protein adsorption [24], which has been recognized as a key element of bioactive behavior [35]. Cell attachment to Si-CAP was mainly mediated by the α5β1 integrin receptor and to a lesser degree by the α2β1 integrin receptor [24]. Taken together, the results of our current study confirmed the excellent osteogenic bioactive properties and resorbability of this bioceramic bone substitute, which were superior to those of β-TCP, in the human case. Consequently, the null hypothesis of our study was accepted on the basis of these results. In addition, the amount of bone formation observed apically in Si-CAP sites was similar to that recorded after SFA using autogenous bone, i.e., the gold standard [36]. A systematic review and meta-analysis of SFA with various bone grafting materials showed that more than 50% of the augmented sites consisted of either residual graft granules or scar tissue rather than vital bone. The use of autologous bone grafts resulted in the greatest extent of de novo bone formation and the lowest amount of residual graft compared to other grafting materials [19]. A 95% survival rate was reported for implants inserted after the bone substitute supported SFA, inducing the formation of tissue which contained 29% vital bone and 25% residual graft material [37], i.e., lower bone formation and graft material resorption than noted for Si-CAP in our study. Our findings are furthermore in agreement with observations by Cadenas-Vacas et al., who reported excellent bone regeneration of extraction sockets 3 months after utilizing Si-CAP for socket preservation in patients. Bone formation was higher with less residual grafting materials being present in sockets grafted with Si-CAP than in sockets in which a bovine-derived hydroxyapatite with well-documented osteoconductive properties was used [38]. Moreover, based on our results, due to its excellent bioactive properties, the Si-CAP bone grafting material appears to be a valuable graft choice in patients with systemic conditions that prevent autograft harvesting or lead to reduced osteoblast activity, such as patients with osteoporosis receiving bisphosphonate medication or patients with other systemic conditions impairing wound healing such as diabetes. Oral bisphosphonate administration can impair bone remodeling and regeneration processes, and thus interfere with grafting procedures, dental implant surgery, and their outcomes as outlined in a recent review by Carossa et al. [39]. In this context, a bone substitute material, which is endowed with a stimulatory effect on osteoblast function and bone tissue formation, appears to be advantageous for bone regeneration procedures in conjunction with dental implant placement in these patient groups with systemic conditions.

In addition to the chemical composition of calcium phosphate bone grafting materials, granule size and porosity affect the amount of bone formation. Previously, we were able to demonstrate that when using β-TCP granules of identical chemical composition and granule size but differing porosity, granules with higher porosity (65% vs. 30%) induced significantly higher de novo bone formation in conjunction with greater β-TCP resorption six months after SFA in humans [3]. With respect to granule size, greater bone formation was noted with bioactive glass 45S5 as well as β-TCP granules of smaller granule size, both in large animals as well as humans [6,40]. In our current study, Si-CAP granules induced significantly higher bone neoformation than β-TCP in the apical region of the biopsies despite displaying a slightly larger granule size than the Ceros^®^ β-TCP granules, while possessing a similar porosity. This is indicative of the superior osteogenic effect of Si-CAP owing to its chemical composition and the surface transformation and ionic dissolution phenomena occurring after contact with biological fluids, which is in excellent agreement with the results of the previous in vitro cell culture as well as the in vivo ovine studies described above [4,24]. The results of our study also corroborate the findings regarding the colonization of Si-CAP scaffolds with combined macro- and microporosity with mesenchymal stem cells. After 7 days of perfusion culture, in vitro Si-CAP scaffolds facilitated homogenous colonization with terminally differentiated osteoblasts and a mineralizing extracellular matrix [28], which then enabled segmental defect repair following scaffold implantation in vivo [41].

With regard to SFA, volume stability of the grafted region over time is an important clinical parameter when evaluating the success of bone substitute materials. Previously, we were able to establish a protocol which allows us to determine volumetric changes in graft volume by superimposing the CBCT DICOM data of the preoperative CBCT, postoperative CBCT, and the CBCT acquired 6 months after sinus floor grafting [6]. This was in addition to using a split mouth design. Moreover, when studying biomaterial-stimulated bone regeneration, gaining insight into the role of angiogenesis has received increasing attention. In this context, we recently were able to advance our hard tissue histologic technique further by developing a resin embedding protocol, which facilitates cutting 5–7 µm thick sections from bioceramic-containing biopsies. As a result, a much larger number of thin sections, in which capillary formation can be visualized at significantly higher resolutions, can be prepared from biopsies only 2.5 mm in diameter compared to using a sawing microtome, which renders seven 50 µm thick sections per biopsy. Using this novel technique, we were able to visualize capillary formation with von Willebrand factor expression in the pores of degrading Si-CAP granules six months after sinus floor grafting (Figure 5i). As a result, a prospective clinical study with a larger patient population has been designed which involves a split-mouth design, investigating angiogenesis and analyzing CBCT data for evaluating the volume stability of the grafted region in addition to detailed histologic analysis of osteogenesis, so as to generate an even more comprehensive dataset to further establish the evidence-based use of Si-CAP in patients. This study will also include a second arm, i.e., patients with implant placement 4 months after SFA.

## 5. Conclusions

Si-CAP granules induced greater and more advanced bone formation and graft material resorption than β-TCP granules, whose remarkable osteoconductivity has been widely demonstrated in patients. Our findings were in agreement with those of previous in vitro and animal studies and confirmed the tested hypothesis. Hence, the detailed histological data generated in this first study in patients are a first step for successful translation to the clinic and evidence-based use of Si-CAP, which consequently appears to be an excellently suited bioactive bone substitute for sinus floor grafting in the clinical arena.

## Figures and Tables

**Figure 1 bioengineering-10-01408-f001:**
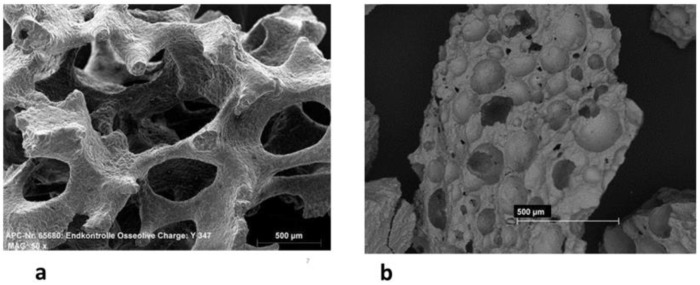
(**a**) Scanning electron micrograph of the Si-CAP (Osseolive^®^) particulate bone grafting material. Bar = 500 µm. (**b**) Scanning electron micrographs of the Ceros^®^ Β-TCP particulate bone grafting material. Bar = 500 µm.

**Figure 2 bioengineering-10-01408-f002:**
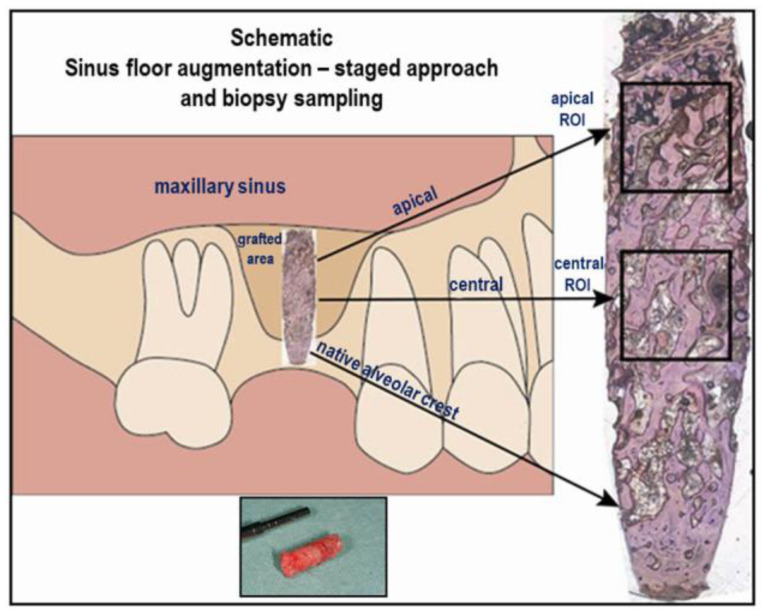
Schematic illustrating the biopsy sampling from patients six months after grafting of the sinus floor utilizing a calcium phosphate bone grafting material as well as the respective histomicrograph with its anatomical orientation and the ROIs.

**Figure 3 bioengineering-10-01408-f003:**
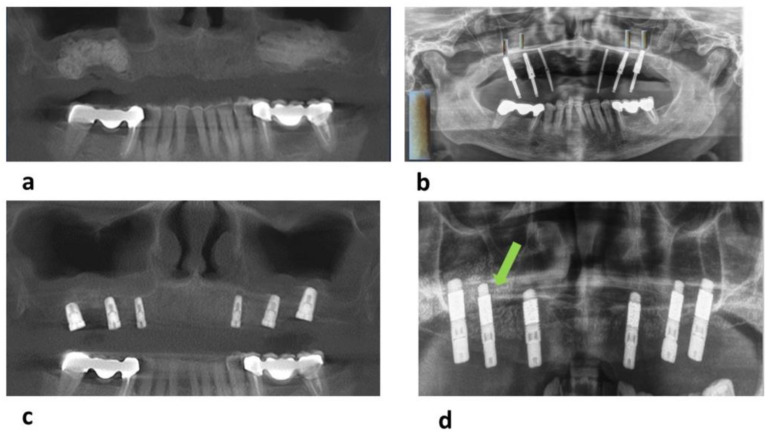
(**a**) Cone beam CT (CBCT) acquired postoperatively in a patient, in whom Si-CAP granules were used for SFA; (**b**) panoramic radiographs taken at preparation of the implant bed 6 months after utilizing Si-CAP granules for sinus floor grafting; (**c**) CBCT acquired subsequent to implant surgery 6 months after bilateral sinus floor grafting using Si-CAP; (**d**) CBCT acquired after implant placement 6 months after SFA with β-TCP granules: residual bone grafting material is clearly visible (green arrow).

**Figure 4 bioengineering-10-01408-f004:**
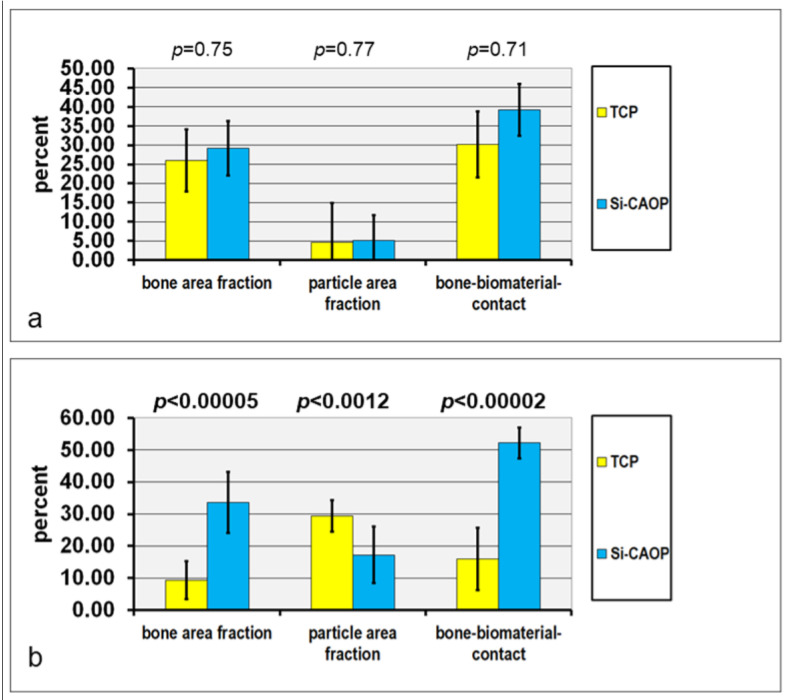
Histograms depicting the results of the histomorphometric analysis: bone area fraction, bioceramic granule area fraction, and bone-bioceramic contact of (**a**) the central ROI and (**b**) the apical ROI in hard tissue sections of either Si-CAP biopsy specimens or β-TCP biopsy samples, which were harvested six months after sinus floor grafting. All values are the mean + standard deviation of 19 measurements.

**Figure 5 bioengineering-10-01408-f005:**
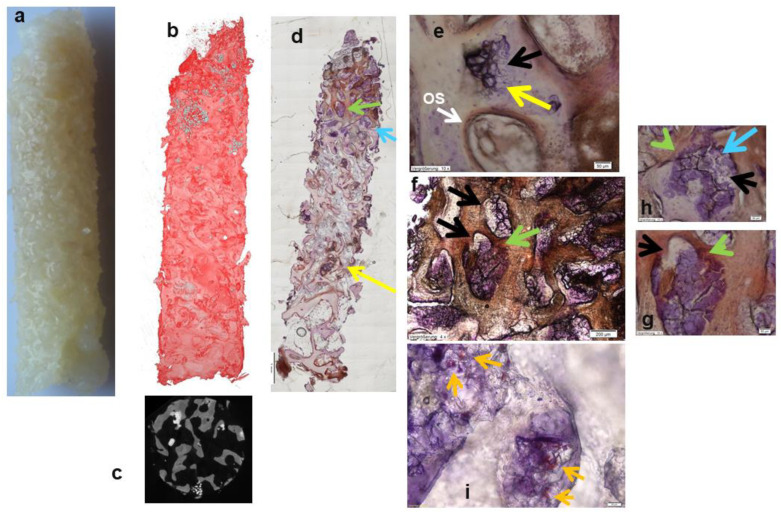
Biopsy sampled 6 months after implanting Si-CAP for sinus floor grafting: (**a**) macroscopic photograph; (**b**,**c**) synchrotron microtomographical image (bone—red; residual grafting materials—grey); (**d**–**h**) histomicrographs of undecalcified hard tissue section with immunodetection of OCN (osteocalcin). Excellent trabecular bone formation is visible after sinus floor grafting. A few highly leached and degraded fragments of the Si-CAP granules are embedded in these newly formed trabeculae. Extensive bone ingrowth into the pores of these residues, which feature excellent bone bonding, is present (**d**,**e**) (yellow arrows); OS—osteoid. At the degrading Si-CAP granule surface, active matrix mineralization and osseous tissue formation with strong OCN expression are visible in the apical region of the biopsies in close proximity to the bordering sinus mucosa (**d**–**g**) (green arrows). As such, progressing bone formation is in tandem with the continuously progressing degradation of the residual Si-CAP granules, which are gradually replaced by the new bone tissue ((**e**–**g**) black arrows; (**d**,**h**) blue arrows); (**i**) histomicrograph of 5 µm undecalcified section immunohistochemically stained for von Willebrand factor: visualization of capillary formation (orange arrows) in bone tissue formed in the pores of a degrading Si-CAP granule six months after sinus floor grafting (bar = 20 µm).

**Figure 6 bioengineering-10-01408-f006:**
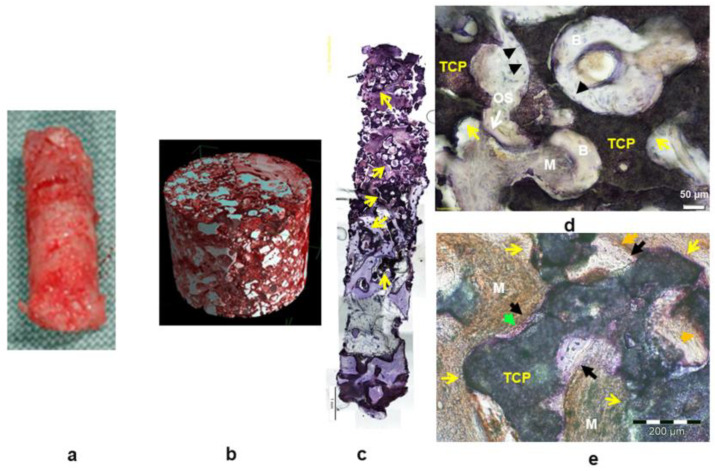
Biopsy harvested 6 months after SFA with β-TCP: (**a**) macroscopic photograph; (**b**) synchrotron microtomographical image (bone—red; residual grafting material—white); (**c**–**e**) histomicrographs of undecalcified hard tissue sections with immunodetection of collagen I (**d**); osteocalcin (**e**). A greater amount of residual β-TCP bone substitute material is visible (**c**–**e**) (yellow arrows) in comparison to Si-CAP biopsy samples. The β-TCP granules display excellent bone-bioceramic contact in the central ROI (**d**) (black arrowheads) with bone ingrowth into the pores of the β-TCP granules. OS—osteoid; B—mineralized bone tissue; M—osteogenic mesenchyme. (**e**) Histomicrograph with β-TCP bone grafting material in the apical ROI. Commencing bone formation (black arrows) at the bioceramic surface (yellow arrows) and osteoblasts with strong staining for OCN (green arrowhead) are present in combination with moderate OCN staining of the osteogenic mesenchyme indicating active matrix mineralization (orange arrows).

**Table 1 bioengineering-10-01408-t001:** Patient clinical data.

Patient No.	Bone Grafting Material	Gender	Age
1	Si-CAP	F	51
2	Si-CAP	F	76
3	Si-CAP	F	69
4	Si-CAP	F	70
5	Si-CAP	F	69
6	Si-CAP	M	68
7	Si-CAP	F	66
8	Si-CAP	M	59
9	Si-CAP	M	58
10	Si-CAP	M	65
11	Si-CAP	F	52
12	Si-CAP	M	70
13	Si-CAP	F	54
14	Si-CAP	M	51
15	Si-CAP	M	59
16	Si-CAP	F	51
17	Si-CAP	M	67
18	Si-CAP	F	64
19	Si-CAP	M	56
20	β-TCP	F	61
21	β-TCP	F	54
22	β-TCP	F	74
23	β-TCP	F	66
24	β-TCP	F	52
25	β-TCP	F	60
26	β-TCP	M	69
27	β-TCP	M	60
28	β-TCP	F	54
29	β-TCP	M	59
30	β-TCP	F	56
31	β-TCP	F	65
32	β-TCP	M	70
33	β-TCP	M	72
34	β-TCP	F	44
35	β-TCP	F	60
36	β-TCP	F	45
37	β-TCP	F	49
38	β-TCP	M	40

(F—female, M—male).

**Table 2 bioengineering-10-01408-t002:** Results of the osteogenic marker expression by immunohistochemical analysis. (a) ALP, BSP, OCN, and type I collagen expression in the cell and matrix components in the apical ROI of Si-CAP or β-TCP biopsies. (b) ALP, BSP, OCN, and type I collagen expression in the cell and matrix components in the central ROI of Si-CAP or β-TCP biopsies.

**a**																			
** Marker **	**Osteoblasts**				**Osteocytes**			**Fibroblastic Cells of the Osteogenic Mesenchym**			**Fibrous Matrix**			**Bone Matrix**			**Osteoid**		
	**Si-CAOP**		**TCP**	** *p* **	**Si-CAOP**	**TCP**	** *p* **	**Si-CAOP**	**TCP**	** *p* **	**Si-CAOP**	**TCP**	** *p* **	**Si-CAOP**	**TCP**	** *p* **	**Si-CAOP**	**TCP**	** *p* **
** ALP **	**0.4 ± 0.2**		**0**	**0.04 ***	**0**	**0**	**0.9**	**0.20 ± 0.1**	**0.3 ± 0.1**	**0.8**	** 2.3 ± 1.5 **	** 3.3 ± 1.3 **	**0.05 ***	** 1.2 ± 1.3 **	**0.8 ± 0.3**	**0.4**	** 1.4 ± 1.6 **	** 1.1 ± 1.5 **	**0.7**
** BSP **	**0.4 ± 0.1**		**0**	**0.04 ***	**0.4 ± 0.15**	**0.1 ± 0.02**	**0.04 ***	**0.5 ± 0.2**	**0**	**0.03 ***	** 3.4 ± 1.2 **	** 2.6 ± 1.4 **	**0.06**	** 2.6 ± 1.3 **	** 1.2 ± 1.4 **	**0.003 ***	** 2.8 ± 1.1 **	** 2.9 ± 1.5 **	**0.2**
** OCN **	** 3.1 ± 1.1 **		**0.2 ± 0.05**	**0.03 ***	**0.3 ± 0.1**	**0.1 ± 0.03**	**0.04 ***	**0.5 ± 0.2**	**0.4 ± 0.1**	**0.06**	** 3.6 ± 1.3 **	** 2 ± 1.8 **	**0.04 ***	** 2 ± 1.7 **	** 2 ± 1.8 **	**0.7**	** 1.3 ± 1.4 **	**0.5 ± 0.2**	**0.03 ***
** ColI **	**0.4 ± 0.15**		**0**	**0.02 ***	**0.2 ± 0.05**	**0.1 ± 0.01**	**0.04 ***	**0.04 ± 0.2**	**0**	**0.9**	** 3.3 ± 1.1 **	** 2.4 ± 1.9 **	**0.1**	** 1 ± 1.2 **	**0.4 ± 0.3**	**0.2**	** 2.5 ± 1.1 **	**0.9 ± 0.1**	**0.0001 ***
**b**																			
** Marker **		**Osteoblasts**			**Osteocytes**			**Fibroblastic Cells of the Osteogenic Mesenchym**			**Fibrous Matrix**			**Bone Matrix**			**Osteoid**		
		**Si-CAOP**	**TCP**	** *p* **	**Si-CAOP**	**TCP**	** *p* **	**Si-CAOP**	**TCP**	** *p* **	**Si-CAOP**	**TCP**	** *p* **	**Si-CAOP**	**TCP**	** *p* **	**Si-CAOP**	**TCP**	** *p* **
** ALP **		**0.6 ± 0.2**	**0**	**0.02 ***	**0.6 ± 0.14**	**0.5 ± 0.3**	**0.99**	**0.6 ± 0.2**	**0.6 ± 0.1**	**0.9**	** 2.8 ± 1.5 **	** 2.6 ± 1.7 **	**0.7**	** 1 ± 1.3 **	** 1.5 ± 1.3 **	**0.2**	** 2.0 ± 1.7 **	** 1.5 ± 1.6 **	**0.2**
** BSP **		**0.4 ± 0.05**	**0.2 ± 0.05**	**0.04 ***	** 1.3 ± 0.7 **	**0.14 ± 0.6**	**0.2**	**0.6 ± 0.1**	**0.2 ± 0.05**	**0.04 ***	** 3.0 ± 1.5 **	** 2.0 ± 1.4 **	**0.02 ***	** 3.5 ± 0.9 **	** 2.6 ± 1.6 **	**0.03 ***	** 3.4 ± 0.7 **	** 3.3 ± 0.9 **	**0.9**
** OCN **		**0.4 ± 0.1**	**0**	**0.02 ***	** 1.2 ± 0.5 **	**0**	**0.25**	**0.6 ± 0.2**	**0**	**0.03 ***	** 1.9 ± 1.4 **	** 1.8 ± 1.6 **	**0.8**	** 3.0 ± 1.4 **	** 3.4 ± 1.4 **	**0.08**	** 1.5 ± 1.4 **	** 2 ± 1.6 **	**0.2**
** Coll I **		**0.4 ± 0.05**	**0.2 ± 0.05**	**0.04 ***	**0.9 ± 02**	**0.9 ± 0.3**	**0.9**	**0.7 ± 0.2**	**0.6 ± 0.3**	**0.9**	** 3.0 ± 1.2 **	** 2.0 ± 1.7 **	**0.2**	** 1.7 ± 1.4 **	**0.8 ± 0.8**	**0.04 ***	** 3 ± 1.3 **	** 1.1 ± 1.1 **	**0.0009 ***

Mean values of the scores ± standard deviation of the osteogenic marker expression (ALP—alkaline phosphatase; BSP—bone sialoprotein; OCN—osteocalcin; Col I—type I collagen), in the different cell and matrix components of the bone tissue. An average score of 3.5–5 was considered to be a strong expression of a respective marker in a given cellular or matrix component, and an average score of (2.3–3.4), (1–2.2), and (0.1–0.9) was a moderate, mild, and minimal expression. All values are the mean ± standard deviation of 19 measurements. Asterisks indicate statistical significance.

## Data Availability

The data presented in this study are available on request from the corresponding author. The data are not publicly available due to privacy restrictions.

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
