# Peer review of "Osteogenic Effect of a Bioactive Calcium Alkali Phosphate Bone Substitute in Humans"

_bioengineering, 2023, doi:10.3390/bioengineering10121408_

Round 1

Reviewer 1 Report

Comments and Suggestions for Authors The work would have been of higher quality if both compared materials had the same porosity and granule size, and if the subjects who needed a bilateral sinus lift were chosen, where each side would be augmented with a different material. In this way, the difference in results due to individual biological response would be avoided.

Author Response

The aspect regarding difference in granule size has been discussed in the discussion  section also that a more extensive prospective clinical study with a larger patient population has been designed which involves a split-mouth design, investigating angiogenesis and analyzing CBCT data for evaluating the volume stability of the grafted region in addition to detailed histologic analysis of osteogenesis, so as to generate an even more comprehensive dataset for further establishing the evidence-based use of Si-CAP in patients. The porosity of both test materials is similar.

Reviewer 2 Report

Comments and Suggestions for Authors

Dear Authors, 

you made a really great work!

The paper is a clinical study on a osteogenic effect of a bioactive calcium alkali phosphate bone

substitute in humans.

The Authors made a great work in terms of methodology and the paper sounds scientific and well written.

However, some improvements are mandatory before acceptance.

The abstract is well written, complete and summary in its various aspects. The keywords are complete and appropriate. Please add a specification of the type of clinical work that was done, and in materials and methods section.

In the introduction:

·        “In this context, detailed histologic analysis of the bone cell and tissue responses to bone substitutes after implantation in vivo can contribute significantly to generating the required knowledge base for evidence-based application of these bone substitutes in humans and for successful translation to the clinic. In this context, we previously developed a hard tissue histologic technique that enables visualizing osteoblasts that actively form and secrete osseous matrix and induce its mineralisation at the bone-bioceramic interface and in the pores of degrading bioactive bone substitutes at varying time points after the grafting procedure [23, 28, 29].”  From this point of view, I think it is interesting to consider some aspects, in particular how to help the cellular colonization of the substrate used for this purpose, which with an important mineral matrix, must be facilitated as a possibility of colonization by cells and blood vessels, such as explained by: “Sayed, M.E.; Mugri, M.H.; Almasri, M.A.; Al-Ahmari, M.M.; Bhandi, S.; Madapusi, T.B.; Varadarajan, S.; Raj, A.T.; Reda, R.; Testarelli, L.; et al. Role of Stem Cells in Augmenting Dental Implant Osseointegration: A Systematic Review. Coatings 2021, 11, 1035. https://doi.org/10.3390/coatings11091035”

Materials and methods are clear and well explained. The authors did a great job in the explanation of all the variables identified and included in the study. The methodology with which the study was carried out is absolutely clear and repeatable.

Results are easy to understand and comprehensive. All the studied characteristics were reported in tables which are clear and concise. The images are really of excellent quality. I suggest the Authors include a comment on the images in the discussion, considering the interesting aspects that emerged from this analysis and the strengths and weaknesses.

The article is overall really well written, effective and complete. The explanations are easy and the authors have done a really good job in describing different aspects of the manuscript.

In the discussion:

·        I suggest that the authors also underline in the discussions the extremely interesting data that emerged in comparison between trapezoidal and parallel threads.

·        The overall is comprehensive, concise and complete in its various aspects.

Conclusions are concise and clear.

Bibliography is formatted respecting the journal’s requirements, updated, and no improper citations are evidenced.

Figures and labels are clear and easy to comprehend.

English is clear and easy to understand.

Author Response

In a previous study we elucidated mechanisms underlying the enhanced effect of attracting osteoprogenitor cells to the Si-CAP bioceramic surface followed by cell signalling which governs cell differentiation and survival. This effect was related to calcium uptake at the top layer of the Si-CAP bioceramic, to silicon-ion release, and increased serum protein adsorption of fibronectin, as well as simultaneous enhanced activation of intracellular signaling that modulates bone cell differentiation as well as survival . This previous study is outlined in the introduction section. p 2

In addition two previous studies  dealt with colonization of Si-CAP scaffolds with mesenchymal stem cells for bone tissue engineering and segmental defect repair. A few lines have been added to the discussion section outlining the respective excellent findings p.12.

The study cohort and study was not geared towards examining the effect of implant thread design on bone implant contact.

Reviewer 3 Report

Comments and Suggestions for Authors

introduction

''The objective of the current study was to correlate the results of previous in vitro cell  culture and in vivo animal studies that analyzed the effect of Si-CAP on intracellular  signaling, osteoblast differentiation, bone matrix maturation and bone tissue formation in vitro and in vivo in clinically representative large animal models in comparison to β-TCP and other CaPs, with the histomorphometric and immunohistochemical findings 106regarding the effect of Si-CAP on osteoblast differentiation, bone tissue maturation and formation six months after sinus floor grafting in humans in comparison to β-TCP.'' 

This is written as a single sentence and therefore is way too long and also confusing. Separate into smaller sentences and rephrase. 

Materials and methods

- Even if the synthesis fabrication and material were described in previous articles, report the main characteristic for both of the group in full. This will help the understanding of the experiment and a more fast reproducibility.

- For the patients selection, the gender and ages should be reported in the results section.

- Described in more details the inclusion/exclusion criteria for patient selection. Be as more accurate as possibile to guarantee the reproducibility of the study

- Provide information on the implants adopted

Discussion

- Discuss if the null hypotheses was accepted or rejected based on the results

- Discuss some example of systemic patients were autologous bone harvest is not indicated and they would beneficial from other form of augmentation, such as patients that are assuming bisphosphonates. 

For this propouse discuss and cite the following article Carossa, M.; Scotti, N.; Alovisi, M.; Catapano, S.; Grande, F.; Corsalini, M.; Ruffino, S.; Pera, F. Management of a Malpractice Dental Implant Case in a Patient with History of Oral Bisphosphonates Intake: A Case Report and Narrative Review of Recent Findings. Prosthesis 20235, 826-839. https://doi.org/10.3390/prosthesis5030058

Author Response

see attachment 

Reviewer’s comments:

introduction

''The objective of the current study was to correlate the results of previous in vitro cell culture and in vivo animal studies that analyzed the effect of Si-CAP on

intracellular signaling, osteoblast differentiation, bone matrix maturation and bone tissue formation in vitro and in vivo in clinically representative large animal models in comparison to β-TCP and other CaPs, with the histomorphometric and immunohistochemical findings 106regarding the effect of Si-CAP on osteoblast differentiation, bone tissue maturation and formation six months after sinus floor grafting in humans in comparison to β-TCP.''

This is written as a single sentence and therefore is way too long and also confusing. Separate into smaller sentences and rephrase.

The respective sentence has been rephrased and divided into shorter sentences.

Materials and methods

-         Even if the synthesis fabrication and material were described in previous articles, report the main characteristic for both of the group in full. This will help the understanding of the experiment and a more fast reproducibility.

Synthesis and fabrication of both bone grafting materials have been described in more detail and there main characteristics have been outlined in full.

-         For the patients selection, the gender and ages should be reported in the results section.

A table outlining age and gender of the patients has been added to the results section.

-         Described in more details the inclusion/exclusion criteria for patient selection. Be as more accurate as possibile to guarantee the reproducibility of the study

The inclusion and exclusion criteria for the patient selection have been described in more detail.

-         Provide information on the implants adopted

Information regarding the implants placed 6 months after sinus floor augmentation has been added.

Discussion

-         Discuss if the null hypotheses was accepted or rejected based on the results

A passage emphasizing that the null hypotheses was accepted based on the results has been added.

-         Discuss some example of systemic patients were autologous bone harvest is not indicated and they would beneficial from other form of augmentation, such as patients that are assuming bisphosphonates.

Examples of patients with systemic conditions, in which autologous bone harvest is not indicated or in which wound healing and osteoblast function is impaired, have been discussed mentioning the possible benefit of osteoblast function enhancing bone substitutes. In this context, the publication mentioned by the reviewer has been cited.

For this propouse discuss and cite the following article Carossa, M.; Scotti, N.; Alovisi, M.; Catapano, S.; Grande, F.; Corsalini, M.; Ruffino, S.; Pera, F. Management of a Malpractice Dental Implant Case in a Patient with History of Oral Bisphosphonates Intake: A Case Report and Narrative Review of Recent Findings. Prosthesis 2023, 5, 826-839. https://doi.org/10.3390/prosthesis5030058

This publication recommended by the reviewer has been cited, reference 39.

Reviewer 4 Report

Comments and Suggestions for Authors

The study is very pertinent and of high clinical interest and can overcome some limitations of bone regeneration.

Abstract

Keywords - Why did the authors write ) at the end of the sentence?

Introduction

Lines 61, 66 – The references are missing

Materials and Methods

Line 171- “…cutting 50 m thick sections ...” -  Is it really like that, 50 m?

Discussion

Lines 354 – the references are missing

References

The references need to be updated - they are very old.

Round 2

Reviewer 3 Report

Comments and Suggestions for Authors

The authors revised the manuscript correctly.